# Concentrations of Plasma Amino Acids and Neurotransmitters in Participants with Functional Gut Disorders and Healthy Controls

**DOI:** 10.3390/metabo13020313

**Published:** 2023-02-20

**Authors:** Shanalee C. James, Karl Fraser, Janine Cooney, Catrin S. Günther, Wayne Young, Richard B. Gearry, Phoebe E. Heenan, Tania Trower, Jacqueline I. Keenan, Nicholas J. Talley, Warren C. McNabb, Nicole C. Roy

**Affiliations:** 1The Riddet Institute, Massey University, 4474 Palmerston North, New Zealand; 2School of Food and Advanced Technology, Massey University, 4472 Palmerston North, New Zealand; 3AgResearch, Tennent Drive, 4442 Palmerston North, New Zealand; 4High-Value Nutrition National Science Challenge, 1023 Auckland, New Zealand; 5The New Zealand Institute for Plant and Food Research Ltd., Ruakura Campus, Bisley Road, 3214 Hamilton, New Zealand; 6Department of Medicine, University of Otago, 8011 Christchurch, New Zealand; 7Department of Surgery, University of Otago, 8011 Christchurch, New Zealand; 8School of Medicine and Public Health, The University of Newcastle, Callaghan, Newcastle 2308, Australia; 9Department of Human Nutrition, University of Otago, 9016 Dunedin, New Zealand

**Keywords:** amino acids, neurotransmitters, irritable bowel syndrome, functional gastrointestinal disorder

## Abstract

Amino acids are important in several biochemical pathways as precursors to neurotransmitters which impact biological processes previously linked to functional gastrointestinal disorders (FGIDs). Dietary protein consumption, metabolic host processes, and the gut microbiome can influence the plasma concentration of amino acids and neurotransmitters, and their uptake by tissues. The aim of this analysis was to quantify 19 proteogenic and 4 non-proteogenic amino acids and 19 neurotransmitters (including precursors and catabolites, herein referred to as neurotransmitters) to ascertain if their circulating concentrations differed between healthy participants and those with FGIDs. Plasma proteogenic and non-proteogenic amino acids and neurotransmitters were measured using ultra-performance liquid chromatography and liquid chromatography–mass spectrometry, respectively, from 165 participants (Rome IV: irritable bowel syndrome (IBS-constipation, IBS-diarrhea), functional constipation, functional diarrhea, and healthy controls). There were significant differences (*p* < 0.05) in pairwise comparisons between healthy controls and specific FGID groups for branched-chain amino acids (BCAAs), ornithine, and alpha-aminobutyric acid. No other significant differences were observed for the neurotransmitters or any other amino acids analyzed. Multivariate and bivariate correlation analyses between proteogenic and non-proteogenic amino acids and neurotransmitters for constipation (constipation (IBS-C and functional constipation) and phenotypes diarrhea (IBS-D and functional diarrhea)) and healthy controls suggested that associations between BCAAs, 5-hydroxytryptophan, and kynurenine in combination with tyrosine, 3,4-dihydroxyphenylalanine, and 3,4-dihydroxyphenylacetic acid and associations with gamma-aminobutyric acid, glutamate, asparagine, and serine are likely disrupted in FGID phenotypes. In conclusion, although correlations were evident between some proteogenic and non-proteogenic amino acids and neurotransmitters, the results showed minor concentration differences in plasma proteogenic and non-proteogenic amino acids, amino acid-derived metabolites, and neurotransmitters between FGID phenotypes and healthy controls.

## 1. Introduction

Amino acids (both proteogenic and non-proteogenic) are important metabolites that are precursors to several crucial metabolites and pathways in functional gastrointestinal disorders (FGIDs). Plasma concentrations of both proteogenic and non-proteogenic amino acids have been reported to differ in diabetes, obesity, inflammatory bowel disease, and metabolic dysfunction compared with healthy controls [1,2,3,4,5]. However, limited data are available for their possible role in FGIDs [6,7]. The importance of proteogenic amino acids is not just in the variation in their circulatory concentrations but also in their conversion to metabolites as precursors of neurotransmitters with important biochemical functions in FGIDs [8,9].

Compared to proteogenic and non-proteogenic amino acids, neurotransmitters are more commonly investigated in irritable bowel syndrome (IBS) due to their link to the gut–brain axis and possible perturbed gut comfort pathways [10,11]. For example, glutamine (GLN), a regulator of intestinal permeability and function [12], and histidine (HIS), the precursor to histamine, have potential importance in immune system responses [13]. In addition, the degradation of aromatic amino acids (tryptophan (TRP), tyrosine (TYR), and phenylalanine (PHE)) can produce neurotransmitters, gamma-aminobutyric acid (GABA), norepinephrine (NE), dopamine (DA), and serotonin (5-HT) [14]. The neurotransmitter 5-HT, for example, is postulated to be important in IBS, with prior research showing links to visceral sensitivity [15,16,17,18]. Understanding how circulating concentrations of proteogenic amino acids, amino acid metabolites (including non-proteogenic amino acids), and neurotransmitters are altered and interrelated in FGIDs could highlight altered pathways in intestinal tissues and elsewhere in the body between individuals with FGIDs and healthy controls.

The aim of this analysis was to quantify the concentration of plasma proteogenic and non-proteogenic amino acids and neurotransmitters (including precursors and catabolites) in a cohort of individuals with FGIDs (irritable bowel syndrome-constipation (IBS-C), IBS-diarrhea (IBS-D), functional constipation (FC), functional diarrhea (FD)) and healthy controls. It was hypothesized that plasma amino acid and neurotransmitter concentrations would differ between FGID phenotypes and healthy controls, reflecting impaired gastrointestinal function in individuals with FGIDs. Therefore, the concentration of 23 (proteogenic and non-proteogenic) amino acids and 19 neurotransmitters were quantified in plasma samples collected from the COMFORT (The Christchurch IBS cohort to investigate mechanisms for gut relief and improved transit) cohort in Christchurch, New Zealand [19].

## 2. Materials and Methods

### 2.1. Participants

Analyses were carried out on The Christchurch IBS cohort to investigate mechanisms for gut relief and improved transit (COMFORT cohort; universal trial number: U1111-1216-6662) cohort as previously described [19,20,21]. Symptomatic cases were individuals with IBS or an FGID diagnosed according to the Rome IV criteria (FC, FD, IBS-C, and IBS-D). Individuals presenting with symptoms of IBS-M were excluded from the current analysis. Symptomatic participants undergoing colonoscopy for symptom investigation or surveillance aged 18–70 years were recruited in Christchurch, New Zealand. Healthy controls were asymptomatic individuals undergoing colonoscopy for surveillance due to a family history of colorectal cancer, personal history, or screening for colorectal cancer or polyps aged 18–70 years. Pregnant women or individuals with a known organic disorder (inflammatory bowel disease, colorectal cancer, or diverticulitis), previous bowel resection, and coeliac disease were excluded from the study. One hundred and sixty-five plasma samples were analyzed from the COMFORT cohort for the subsequent amino acid and neurotransmitter analysis. The study was approved by the University of Otago Human Ethics Committee (Ref. # H16/094).

### 2.2. Diet Record and Sample Collection

Dietary records were kept for three sequential days (including one day on the weekend) before plasma collection. Samples were transported to AgResearch, Palmerston North, New Zealand, on dry ice for plasma amino acid analysis. At baseline, blood was drawn in sequential order of 6 × 6 mL lithium heparin (LiH) vacutainer, 3 × 4 mL ethylenediaminetetraacetic acid (EDTA) vacutainer, and 1 × 10 mL untreated vacutainer. Samples were transported to the laboratory and spun within 60 min at 2000× *g* for 5 min at room temperature before being aliquoted into 1.5 mL Eppendorf tubes.

### 2.3. Proteogenic and Non-Proteogenic Amino Acid Analysis

A tungstate precipitation protocol to analyze free proteogenic and non-proteogenic amino acids in EDTA plasma samples was carried out as described in Milan et al., 2015 [22] using ultra-performance liquid chromatography (UPLC) analysis on a Thermo Scientific Dionex Ultimate 3000 (Thermo Scientific, Dornierstrasse, Germering, Germany). Sulfuric acid containing 15 µM L-nor-valine was used as an internal standard for data quantification and analyte recovery with a sodium tungstate protocol and fluorescence derivatization. Data were captured using Chromeleon 7.1 software (Thermo Scientific, Auckland, New Zealand). Standard curves were formulated for each compound within the physiological range of human plasma.

### 2.4. Neurotransmitter Analysis

Inhibitory and excitatory neurotransmitters from the TYR, TRP, and GLU metabolic pathways thought to be involved in the gut–brain axis were measured in Li-Hep plasma (100 µL) using a mass spectrometry probe (MS-probe) and stable isotope coding liquid chromatography–mass spectrometry (LC–MS) method as previously described (Parkar et al. 2020). This method uses ^1^H/^2^H_6_-acetic anhydride and ^1^H/^2^H_2_ -2,2,2-trifluoroethanol to quantitatively convert the neurotransmitters to their corresponding acetate or ester to increase their analysis sensitivity. In addition, isotope label coding is enabled using d_6_-acetic anhydride and d_2_-2,2,2-trifluoroethanol to create an internal standard (IS X-DP) for each neurotransmitter (Appendix A).

### 2.5. Statistical Analysis

All analyses were conducted using R statistical software version 4.0.5 (R, Core team 2018). The effect of gender on metabolite composition was investigated for the control group using permanova ([23], 1000 permutations) based on Euclidean distances between variables as implemented in the R package ‘vegan’ [24]. The Shapiro–Wilk test and quantile–quantile probability plots were used to inspect the distribution of data populations using the ‘MVN’ package [25]. As normality was rejected for all but one compound, non-parametric rank sum testing was applied according to Kruskal–Wallis and Mann–Whitney/Wilcoxon at α = 0.05. The package ‘ggpubr’ [26] was used for visualizing the median and quartile groups of participant cohorts as boxplots. Multivariate analysis of correlations was employed using rCCA [27] for exploring relationships between proteogenic and non-proteogenic amino acid and neurotransmitter data, and clustered image maps (CIM) [28] were computed using a hierarchical clustering approach as implemented in ‘Biocmanager-mixomics’ [29] by the Omics Data Integration Project. Spearman’s rank sum correlation test was used for quantifying associations between proteogenic and non-proteogenic amino acid and neurotransmitter data, and significance was tested (α = 0.05) using the false discovery rate correction for multiple comparisons.

## 3. Results

One hundred sixty-five plasma samples were analyzed from individuals of the COMFORT cohort. Symptom questionnaires based on the Rome IV criteria symptom questionnaire clustered participants as FC n = 25, FD n = 8, IBS-C n = 20, IBS-D n = 40, and healthy control n = 72 (Table 1).

Analysis of dietary protein information (total three-day dietary intake) for participants of all groups (n = 165) showed a significant difference (*p* = 0.043) of approximately 10 mg per day higher intake in males compared to females (Figure 1A). However, there was no significant difference in intake between any groups (*p* = 0.80) (Figure 1B). 

### 3.1. Plasma Amino Acid and Neurotransmitter Concentrations

The plasma concentration of proteogenic and non-proteogenic amino acids and neurotransmitters analyzed is presented in Table 2. Univariate analyses showed no significant differences (*p* > 0.05) between the groups for the 23 circulating proteogenic and non-proteogenic amino acids and 19 neurotransmitters analyzed. The total concentration of proteogenic and non-proteogenic amino acids was not significantly different (*p* > 0.05) between the groups, nor was the total concentration of neurotransmitters (Appendix A). There was a significant decrease (*p* = 0.02) in BCAA concentration in the IBS-C group (415.3 ± 117.27 µM) compared to healthy controls (480.5 ± 125.60 µM) (Figure 2A). The concentration of large neutral amino acids (LNAAs), non-essential amino acids (NEAAs), and essential amino acids (EAAs) were similar (*p* > 0.05) between the groups.

The concentration of aromatic amino acids PHE, TRP, and TYR did not significantly differ (*p* > 0.05) between the groups. TRP, the precursor to 5-HT, kynurenine (KYN), 5-hydroxyindolacetic acid (5-HIAA), and 5-hydroxytryptophan (5-HTP), was not significantly different (*p* > 0.05) in concentration between the groups, with an average concentration of 46.2 ± 15.31 nM. KYN had the highest concentration among metabolites measured, which was similar (*p* > 0.05) between the groups (average 8098.1 ± 2051.79 nM). The concentration of 5-HT metabolites was also similar (*p* > 0.05) between the groups. The 5-HT concentration was, on average, 510.1 ± 512.64 nM, which was approximately 3-fold higher than 5-HIAA and 10-fold higher than 5-HTP.

The concentration of TYR was similar (*p* > 0.05) between the groups and on average 73.5 ± 15.34 µM. Plasma levels of L-DOPAC, HVA, and 3-MT were also similar (*p* > 0.05) between the groups, with the highest concentrations detected for L-DOPAC (126.3 ± 89.85 nM) and HVA (143.4 ± 70.01 nM) but only trace amounts of 3-MT. These observations suggest that L-DOPAC was the primary intermediate for HVA in the TYR metabolic pathway, regardless of group.

Concentrations of DA, an important neurotransmitter, and derived metabolites (NE), epinephrine (E), normetanephrine (NM), and 3,4-dihydroxyphenylglycol (DHPG) were similar (*p* > 0.05) between the groups. Their concentrations were in the low nM range, while that of 3,4-dihydroxyphenylglycol (DHPG) and its derivative 3-methoxy-4-hydroxyphenylglycol (MHPG) were three- and ten-fold higher, respectively. Vanillylmandelic acid (VMA), a metabolite derived from NM or metanephrine (MN), had a similar concentration (*p* > 0.05) between the groups (average 61.61 ± 20.54 nM) with only traces of MN detected.

The plasma concentration of GLN was similar (*p* > 0.05) between the groups. As expected, GLN was the most abundant proteogenic amino acid measured, and its concentration averaged 597.4 ± 69.95 µM across the groups. Glutamic acid (GLU), the precursor to alpha- and gamma-aminobutyric acid (AABA and GABA), did not significantly differ (*p* > 0.05) in concentration between the groups and was found at an average concentration of 52.0 ± 23.54 µM. GABA concentration was similar (*p* > 0.05) between the groups and, on average, 777.1 ± 284.25 µM, which was a five-fold higher concentration than AABA. There was a 20% lower AABA concentration (*p* = 0.023) between the healthy control (138.6 ± 57.90 nM) and IBS-D (114.6 ± 42.94 nM) groups, but this was not observed in FD or other FGID groups compared to healthy controls (Figure 2B).

There was a significant concentration difference in ornithine (ORN) (*p* = 0.038) between healthy controls (62.7 ± 22.92 µM) and IBS-C (50.5 ± 16.02 µM) but not when compared to FC (63.4 ± 14.28 µM). However, no significant differences (*p* > 0.05) were observed in metabolites closely related to ORN, arginine (ARG), and citrulline (CIT).

### 3.2. Correlations between Amino Acids and Neurotransmitters

A multivariate approach was taken using regularized canonical correlation analysis (rCCA) to visually identify probable biologically relevant associations between plasma concentrations of neurotransmitters and amino acids in the healthy control and FGID groups (Figure 3). The strength and significance of pairwise associations were then evaluated using Spearman’s rank sum correlation test (Appendix A). Subtypes were combined into symptomized groups of constipation (FC and IBS-C) and diarrhea (FD and IBS-D) for these analyses.

Using multivariate data exploration, a cluster of positively associated proteogenic and non-proteogenic amino acids and neurotransmitters was identified in the control group (Figure 3A), including TYR and its derived DA-linked neurotransmitters L-DOPA and L-DOPAC. BCAAs were also part of this cluster and associated with the 5-HT-related metabolites, 5-HTP and KYN. Bivariate correlation analysis showed that L-DOPA and L-DOPAC were significantly (Rho = 0.26, *p* < 0.03, Appendix A) positively correlated with TYR in the control group. BCAAs were significantly (*p* < 0.01) positively correlated with 5-HTP and KYN in the control group. Meanwhile, although 5-HT was weakly negatively (not statistically significant) associated with these amino acids, positive associations were identified with taurine (TAU) (Rho = 0.43, *p* < 0.01), aspartic acid (ASP) (Rho = 0.29, *p* < 0.01), serine (SER) (Rho = 0.29, *p* < 0.2), and asparagine (ASN) (Rho = 0.27 *p* < 0.02) in the control group. GABA was positively correlated with its precursor GLU (Rho = 0.29, *p* < 0.02), proline (PRO) (Rho = 0.26, *p* < 0.03), and ASP (Rho = 0.31, *p* < 0.01).

As apparent from Figure 3, associations between proteogenic and non-proteogenic amino acids and neurotransmitters were largely different in the constipation (Figure 3B) and the diarrhea groups (Figure 3C) compared with the control group (Figure 3A). In common with the control group, 5-HT was also significantly positively correlated with ASP and TAU (Rho = 0.34–0.46, *p* < 0.02); however, 5-HTP and KYN were not significantly correlated with TYR or BCAAs. In agreement with the bivariate analyses, TYR was positively associated with L-DOPA (Rho = 0.33–0.36, *p* < 0.03) and weakly associated with L-DOPAC (Rho = 0.02–0.14) in the diarrhea and constipation groups.

In contrast to rCCA, bivariate Spearman correlation analysis identified positive correlations for DA with BCAAs (Rho = 0.31–0.33, *p* < 0.04) in the constipation group, which was not observed in other cohorts. In common with the control group, GABA was positively correlated with PRO in the FGID groups (Rho = 0.27–0.38, *p* < 0.06), but not with ASP or GLU. In the diarrhea group, GABA was positively correlated to SER (Rho = 0.31, *p* = 0.03) and ASN (Rho = 0.29, *p* = 0.05). In contrast to the control group, these amino acids showed a slightly negative although not statistically significant association with 5-HT in pairwise correlation testing. Although the 5-HT precursor TRP showed a trend for negative associations with GABA in all cohorts, this was only statistically significant for the diarrhea group (*p* = 0.01).

## 4. Discussion

This study is the first to report quantitative data on the plasma concentration of 23 proteogenic and non-proteogenic amino acids and 19 neurotransmitters in participants with FGIDs and healthy controls. Contrary to the stated hypothesis, the data showed minimal differences in the concentration of individual amino acids, EAA, LNAAs, NEAAs, and neurotransmitters, except for BCAAs, AABA, and ORN, between healthy controls and FGID subtypes.

In line with similar findings [6,30], there were no differences in plasma concentration of TRP and TRP neurotransmitter products between IBS subtypes, functional groups, or healthy controls. TRP is a precursor of 5-HT, KYN, 5-HIAA, and 5-HTP and is thought to play a critical role in IBS symptomology due to the effect of these neurotransmitters on disrupting motility, sensitivity, and secretion within the gastrointestinal tract [9]. Although prior studies have noted differences in plasma abundance of 5-HT and 5-HT metabolites, there is no conclusive finding between FGID subtypes across studies [31,32,33,34].

Unexpectedly, there were no differences in plasma GLN concentration between FGID subtypes and healthy controls. GLN is recognized as an important regulator of intestinal permeability and function [12,35], as shown by membrane component destruction, altered tight junction distribution, and increased inflammation in an in vitro model of the intestinal epithelium and colonic tissue taken from FGID individuals [12,36]. In another study, supplementation of GLN to individuals with FGIDs decreased concentrations of the pro-inflammatory cytokines interleukin-6 and interleukin-8 compared to healthy controls suggesting that GLN could be a modulator for inflammatory markers in the intestine [37].

Exploring metabolite–metabolite interactions showed positive associations of BCAAs and NEAAs with neurotransmitters 5-HT, DA, and GABA, suggesting that the metabolism of these neurotransmitters was altered in participants with constipation and diarrhea-predominant phenotypes when compared to healthy controls. TYR and TRP, part of the LNAA group, compete with BCAAs for transport across the blood–brain barrier via the L-amino acid carrier. Therefore, changes in plasma BCAA concentration can modify TRP and TYR uptake by the brain and their subsequent metabolism to 5-HT and catecholamines such as DA and their catabolites, respectively [38]. Although concentrations of TRP neurotransmitter products (5-HT, KYN, 5-HIAA, and 5-HTP) were comparable between groups, their associations with BCAAs and LNAAs differed. In healthy individuals, 5-HT was negatively correlated with BCAAs and LNAAs, in contrast to 5-HTP, KYN, L-DOPA, and L-DOPAC, which had positive correlations with these amino acids. There was no association between these metabolites in constipation or diarrhea-predominant groups. While the underlying mechanisms of such relationships remain to be elucidated, a pairing of 5-HT with BCAAs and LNAAs suggests that these metabolites might affect metabolism interdependently and impact 5-HT uptake in the brain.

There are some limitations related to dietary records, sample collection, and sample analysis. The accuracy of the diet dataset relies on the participants accurately recording their dietary intake. Amino acids and neurotransmitters are circulatory metabolites influenced by dietary intake as well as host and microbial metabolism, and these factors can impact the findings, and may not reflect intestinal concentrations. There is also potential variation in sample collection and aliquots, which could alter the results. The variation in analytical techniques, cohort size, and phenotype grouping can also make comparisons between studies difficult. Metabolite measurements were characterized by high standard errors (far greater than those derived due to the analytical methodology), which we postulate is due to the high variability between participants when attempting to group individuals into defined FGID subtypes. The small sample size of the FD group could present a limitation to statistical inferences; however, when this group was combined into a generalized diarrhea group with IBS-D, this was comparable to the size of the constipation group. The cohort was predominantly female characteristic across all groups and reflective of worldwide FGID populations; however, this could present a limitation to the findings from this study as we were not able to test for the effect of gender in the FGID groups.

## 5. Conclusions

The findings reported here are the first investigation of a comprehensive panel of proteogenic and non-proteogenic amino acids and neurotransmitters in a large FGID cohort. Overall, there were minor differences in the plasma concentration of proteogenic and non-proteogenic amino acids (BCAAs) and neurotransmitters (AABA and ORN) in plasma samples between specific FGID groups and healthy controls. Therefore, no metabolites measured here are considered feasible as robust biomarkers for distinguishing between healthy controls and those with FGIDs, or between FGID groups. Although minor differences were evident between FGID groups, the comprehensive panel of neurotransmitters and proteogenic and non-proteogenic amino acids profiled in this study are nonetheless significant in progressing this field and the underlying mechanisms of the differential relationships between plasma 5-HT, BCAAs, and LNAAs across FGID and control groups are worth further investigation.

## Figures and Tables

**Figure 1 metabolites-13-00313-f001:**
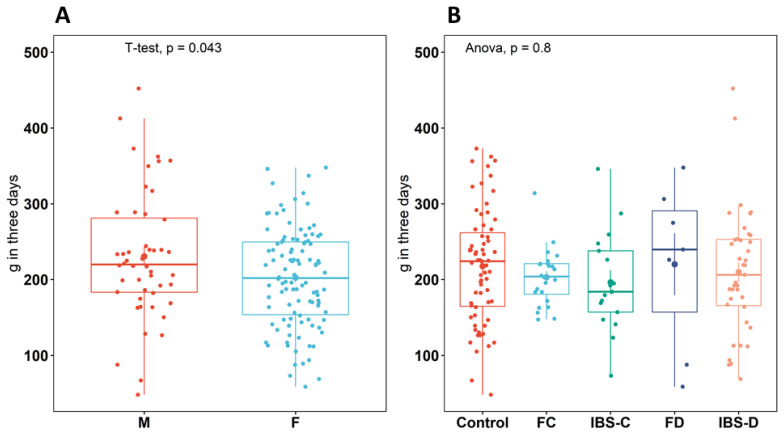
Total sum (g) of dietary protein intake over three days recording using diet diaries for each participant between (**A**): male and female individuals (**B**): between healthy control (control), IBS subtypes (IBS-C, IBS-D), and functional groups (FC, FD), data from males and females combined. Boxplots show the median (center line), 25th and 75th percentile (top and bottom of boxes, respectively), and whiskers representing 1.5 times the interquartile range. Abbreviations: healthy control (control), functional constipation (FC), IBS-constipation (IBS-C), functional diarrhea (FD), and IBS-diarrhea (IBS-D). Colors identifies subtype or healthy control groups.

**Figure 2 metabolites-13-00313-f002:**
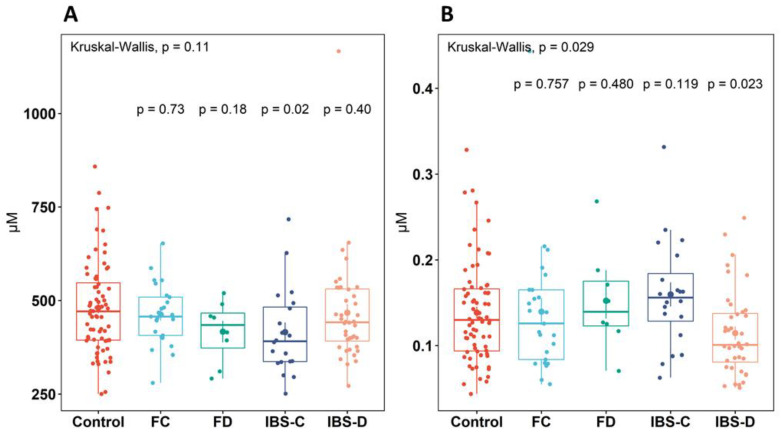
Significant differences found in (**A**): concentration of branched-chain amino acids (BCAAs) and (**B**): alpha-aminobutyric acid (AABA) between healthy control, IBS subtypes, and functional groups. The top significance value denotes the overall significance between all groups. Individual group significance values are pairwise comparisons between healthy controls and the group. Boxplots show the median (center line), 25th and 75th percentile (top and bottom of boxes, respectively), and whiskers representing 1.5 times the interquartile range. Abbreviations: healthy control (control), functional constipation (FC), IBS-constipation (IBS-C), functional diarrhea (FD), IBS-diarrhea (IBS-D). Colors identifies subtype or healthy control groups.

**Figure 3 metabolites-13-00313-f003:**
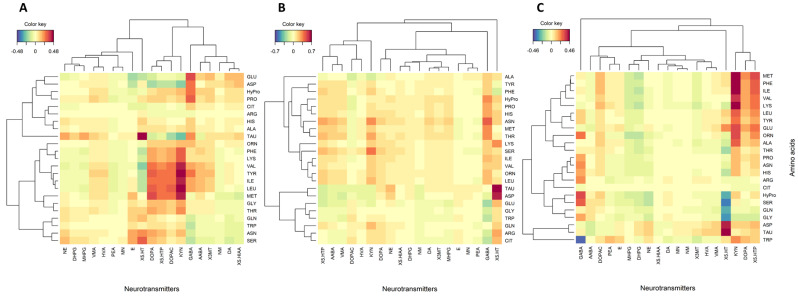
Clustered image maps (no cutoff) between plasma neurotransmitters and proteogenic and non-proteogenic amino acids for (**A**): healthy control, (**B**): constipation group (FC+IBS-C), and (**C**): diarrhea group (FD+IBS-D).

**Table 1 metabolites-13-00313-t001:** Participant number and gender. Abbreviations: healthy control, functional constipation (FC), IBS-constipation (IBS-C), functional diarrhea (FD), and IBS-diarrhea (IBS-D).

	Healthy Control	FC	FD	IBS-C	IBS-D
Female count	37	17	7	19	32
Male count	35	8	1	1	8

**Table 2 metabolites-13-00313-t002:** Concentration of plasma amino acid and neurotransmitter metabolites. Mean values ± standard deviation are presented as either nM or μM of plasma. Abbreviations: Healthy control (control), functional constipation (FC), IBS-constipation (IBS-C), functional diarrhea (FD), IBS-diarrhea (IBS-D), branch chain amino acids (BCAAs), essential amino acids (EAAs), non-essential amino acids (NEAAs), large neutral amino acids (LNAAs).

Metabolite	Healthy Control	FC	FD	IBS-C	IBS-D
Neurotransmitters, nM					
Phenethylamine (PEA)	2.0 ± 1.11	2.5 ± 1.51	2.0 ± 0.63	1.7 ± 0.68	2.1 ± 0.96
3,4-Dihydroxyphenylalanine (L-DOPA)	28.5 ± 11.12	25.5 ± 9.11	25.3 ± 5.16	27.7 ± 7.61	26.6 ± 9.05
Dopamine (DA)	4.4 ± 7.61	5.5 ± 8.96	14.9 ± 29.09	5.9 ± 7.92	4.7 ± 6.49
3-Methoxytyramine (3-MT)	0.6 ± 0.38	0.7 ± 0.56	1.1 ± 1.07	0.6 ± 0.64	0.6 ± 0.67
3,4-Dihydroxyphenylacetic acid (L-DOPAC)	103.3 ± 65.42	116.2 ± 99.20	120.9 ± 38.60	165.4 ± 142.19	125.8 ± 103.86
Homovanillic acid (HVA)	140.5 ± 60.87	131.5 ± 72.71	158.5 ± 88.98	136.9 ± 65.50	149.8 ± 67
Norepinephrine (NE)	3.7 ± 1.34	3.7 ± 1.95	4.7 ± 2.08	3.7 ± 1.49	3.6 ± 1.48
3,4-Dihydroxyphenylglycol (DHPG)	10.9 ± 3.00	10.9 ± 4.01	11.4 ± 3.44	11.2 ± 3.53	10.8 ± 2.95
3-Methoxy-4-hydroxyphenylglycol (MHPG)	34.3 ± 7.19	33.4 ± 10.25	36.6 ± 7.83	33.2 ± 7.93	33.1 ± 7.75
Normetanephrine (NM)	2.1 ± 0.61	2.5 ± 1.43	2.3 ± 0.39	2.0 ± 0.83	2.0 ± 0.68
Epinephrine (E)	0.3 ± 0.20	0.2 ± 0.13	0.2 ± 0.11	0.2 ± 0.14	0.2 ± 0.21
Metanephrine (MN)	0.8 ± 0.20	0.8 ± 0.26	0.8 ± 0.18	0.7 ± 0.17	0.7 ± 0.17
Vanillylmandelic acid (VMA)	61.4 ± 18.72	63.2 ± 18.31	64.2 ± 22.9	59.9 ± 22.81	59.4 ± 19.98
Kynurenine (KYN)	8604.6 ± 2727.89	7700.1 ± 1601.01	7434.1 ± 2017.47	8439.3 ± 1392.45	8312.5 ± 2520.13
5-Hydroxytryptophan (5-HTP)	49.3 ± 18.44	46.9 ± 23.27	45.7 ± 17.18	57.0 ± 21.24	48.0 ± 16.57
Serotonin (5-HT)	585.7 ± 565.47	544.0 ± 446.18	318.8 ± 377.10	620.6 ± 748.40	481.0 ± 426.04
5-Hydroxyindolacetic acid (5-HIAA)	161.0 ± 125.69	176.7 ± 146.02	148.7 ± 111.75	166.9 ± 131.64	213.0 ± 220.91
Alpha-aminobutyric acid (AABA)	138.6 ± 57.90	139.7 ± 78.24	152.3 ± 58.71	159.7 ± 62.91	114.6 ± 42.94
Gamma-aminobutyric acid (GABA)	747.0 ± 230.49	746.1 ± 266.11	839.2 ± 321.81	794.2 ± 321.22	758.7 ± 281.63
Amino acids (including non-proteogenic), μM					
Aspartic acid (ASP)	6.0 ± 3.71	6.2 ± 3.31	6.1 ± 2.66	6.2 ± 3.90	6.1 ± 2.67
Glutamic acid (GLU)	54.5 ± 22.33	57.5 ± 24.42	43.5 ± 16.65	46.6 ± 25.90	58.1 ± 28.40
Hydroxy-proline (HyPRO) *	13.1 ± 5.15	11.4 ± 3.38	14.3 ± 3.10	11.9 ± 4.19	12.7 ± 3.94
Serine (SER)	113.3 ± 22.47	115.0 ± 19.25	124.5 ± 25.23	119.6 ± 23.75	116.4 ± 25.22
Asparagine (ASN)	56.7 ± 12.41	53.8 ± 10.78	54.4 ± 7.42	60.0 ± 10.58	56.6 ± 10.43
Glycine (GLY)	249.0 ± 60.87	258.7 ± 74.34	303.0 ± 90.04	246.4 ± 69.17	259.4 ± 58.67
Glutamine (GLN)	585.5 ± 77.60	607.7 ± 69.08	607.8 ± 49.99	584.3 ± 83.22	601.9 ± 69.84
Histidine (HIS)	59.3 ± 9.70	57.2 ± 7.91	58.3 ± 8.02	60.5 ± 8.91	61.5 ± 10.26
Taurine (TAU) *^,^ **	127.2 ± 44.00	133.7 ± 51.34	148.9 ± 62.27	140.9 ± 54.92	140.3 ± 49.42
Citrulline (CIT) *	35.3 ± 7.99	37.6 ± 10.54	35.0 ± 5.22	35.7 ± 8.56	35.0 ± 8.05
Threonine (THR)	112.0 ± 23.80	116.4 ± 20.89	115.9 ± 18.48	125.9 ± 33.50	116.2 ± 29.05
Alanine (ALA)	402.5 ± 105.12	409.2 ± 87.15	392.1 ± 98.77	370.4 ± 85.40	425.3 ± 114.54
Arginine (ARG)	69.2 ± 16.73	68.8 ± 16.55	80.6 ± 21.49	68.6 ± 21.36	73.5 ± 21.88
Proline (PRO)	281.0 ± 90.15	291.8 ± 100.07	306.1 ± 89.29	249.8 ± 73.52	278.1 ± 107.55
Tyrosine (TYR)	73.8 ± 15.52	80.8 ± 18.78	67.4 ± 7.26	69.6 ± 16.99	75.9 ± 18.11
Valine (VAL)	255.3 ± 54.75	248.9 ± 38.53	224.4 ± 33.84	226.6 ± 53.71	251.7 ± 58.85
Methionine (MET)	31.1 ± 6.71	31.9 ± 5.45	28.4 ± 3.07	29.7 ± 7.69	30.6 ± 6.51
Isoleucine (ILE)	80.8 ± 25.83	75.6 ± 16.43	67.6 ± 18.93	67.9 ± 23.31	78.2 ± 30.87
Leucine (LEU)	144.4 ± 45.03	137.8 ± 28.95	124.5 ± 31.52	120.8 ± 40.24	137.5 ± 57.43
Ornithine (ORN) *	62.7 ± 22.92	63.4 ± 14.28	60.3 ± 17.81	50.5 ± 16.02	63.2 ± 19.51
Phenylalanine (PHE)	66.6 ± 9.73	66.2 ± 8.90	60.4 ± 7.22	64.5 ± 13.01	66.5 ± 10.58
Lysine (LYS)	94.8 ± 22.37	95.3 ± 19.46	96.5 ± 11.56	91.0 ± 22.23	94.0 ± 21.59
Tryptophan (TRP)	47.6 ± 15.86	46.7 ± 12.28	44.4 ± 16.30	46.5 ± 14.50	45.9 ± 17.62
BCAAs (ILE, LEU, VAL)	480.5 ± 125.60	462.3 ± 83.91	416.5 ± 84.30	415.3 ± 117.27	467.4 ± 147.15
EAAs (HIS, THR, VAL, MET, ILE, LEU, PHE, LYS, TRP)	891.8 ± 213.76	876.0 ± 158.80	820.4 ± 148.94	833.4 ± 217.11	882.1 ± 242.78
NEAAs (ALA, ARG, ASN, ASP, GLU, GLN, GLY, PRO, SER, TYR)	1891.6 ± 426.93	1949.4 ± 423.72	1985.4 ± 408.80	1821.3 ± 413.79	1951.3 ± 457.32
LNAAs (HIS, THR, VAL, MET, ILE, LEU, PHE, TYR, TRP)	870.8 ± 206.92	861.4 ± 158.13	791.3 ± 144.65	812.0 ± 211.88	864.0 ± 239.30

Footnote 1: * non-proteogenic amino acids, ** also classified as a neurotransmitter.

## Data Availability

Data will be made available on request. The data are not publicly available due to required packages needed to view raw data.

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
