# Peer review of "Concentrations of Plasma Amino Acids and Neurotransmitters in Participants with Functional Gut Disorders and Healthy Controls"

_metabolites, 2023, doi:10.3390/metabo13020313_

Round 1

Reviewer 1 Report

In the reviewed manuscript, the authors attempted to answer the question of whether functional bowel disorders, such as irritable bowel syndrome - constipation, IBS - diarrhea, functional constipation, or functional diarrhea, affect plasma concentrations of proteogenic and non-proteogenic amino acids and neurotransmitters. The authors analyzed selected amino acids and neurotransmitters in the plasma of 165 participants in the COMFORT cohort study, who were divided into a control group and groups representing the functional bowel disorders. The authors showed only minor differences between the phenotypes studied and the control group, as well as correlations between some proteogenic and non-proteogenic amino acids and neurotransmitters. Several limitations of this study were honestly mentioned and fairly explained by the authors themselves, and these like dietary records, cohort size and variability between participant groups may have had a significant impact on the results obtained and their analysis. There are a few flaws to consider before accepting:

1.      The discussion lacks a reliably formulated conclusion, an attempt of which is present in the abstract section.

2.      The Results section contains a repetition of the data in Table 2. Wasn't it simpler to highlight the significant differences in Table 2 and then just briefly list them in the text? Was it necessary to detail non-significant differences in the text at P>0.05? The authors should rethink their description of the results, making it more concise and simpler. Most of the text from page 7 could simply be shown in a table and thus shortened considerably.

3.      Lines 138-141 are not the results.

Reviewer 2 Report

1.The authors are very often using “19 neurotransmitters”, but in reality they are not measuring “19 neurotransmitters”. They are measuring neurotransmitters, their precursors and products of neurotransmitter breakdown. For the reader not familiar with neurobiology it will be misleading. It should be changed throughout the whole manuscript. May be it could be named “Neurotransmitters metabolome”?

2. Taurine – this compound is listed as a non-proteinogenic amino acid. But it also well known neurotransmitter (see for review Wu and Prentice Journal of Biomedical Science 2010, 17(Suppl 1):S1). As in the paper there is a special emphasis on neurotransmitters – it should be also mentioned from this point of view

3. In the Summary:

 Amino acids are important in several biochemical pathways as precursors to neurotransmitters which impact biological processes previously linked to functional gastrointestinal disorders  (FGIDs), for example, the metabolism of tryptophan to serotonin or conversion of histidine to histamine.

My suggestion is to remove “ for example, the metabolism of tryptophan to serotonin or conversion of histidine to histamine”

4. In the “Discussion”  starting from line 262 authors are commenting on the limitations of their study. I think that authors also should say few words about gender. The effect of gender on metabolite composition was evaluated by special statistical approaches. In spite of it, the great disproportion in the groups (e.g. 19 females and 1 male in the analyzed  IBS-C  group) requires comments on this issue.

Reviewer 3 Report

The manuscript by James et al. studies the plasma of male and female patients with and without functional gastrointestinal disorders (FGID), by comparing the absolute concentrations of 19 Neurotransmitters and 23 amino acids, some of them precursors of the neurotransmitters themselves.

The total number of patients enrolled seems adequate (165). Gender could represent a strong confounding factor, as the study design is clearly unbalanced towards female patients, but this problem has been adequately considered, when focusing on healthy patients.

Healthy subjects have been pairwise compared with the various types of FGID on a molecule-by-molecule basis. Non-normality has been taken correctly into account across all the statistics applied.

An issue appears to be identified in the fact that the possibility of overestimation of p-values due to multiple comparisons has only been addressed for some univariate statistics. The importance of this issue could be classified as minor, because the authors strengthen their observations, by studying the molecules from several perspectives, mixing uni- and multi-variate statistics.

Overall, the manuscript does not evidence strikingly new findings, because it deals only minor concentration differences in plasma proteogenic and non-proteogenic amino acids. This limit, if we can consider it a limit, is explicitly cited in the discussion. Therefore, overall this is an interesting manuscript to read.
